# Human Papillomavirus Same Genotype Persistence and Risk of Cervical Intraepithelial Neoplasia2+ Recurrence

**DOI:** 10.3390/cancers13153664

**Published:** 2021-07-21

**Authors:** Anna Daniela Iacobone, Davide Radice, Maria Teresa Sandri, Eleonora Petra Preti, Maria Elena Guerrieri, Ailyn Mariela Vidal Urbinati, Ida Pino, Dorella Franchi, Rita Passerini, Fabio Bottari

**Affiliations:** 1Preventive Gynecology Unit, European Institute of Oncology IRCCS, 20141 Milan, Italy; eleonora.preti@ieo.it (E.P.P.); mariaelena.guerrieri@ieo.it (M.E.G.); ailyn.vidalurbinati@ieo.it (A.M.V.U.); ida.pino@ieo.it (I.P.); dorella.franchi@ieo.it (D.F.); 2Department of Biomedical Sciences, University of Sassari, 07100 Sassari, Italy; fabio.bottari@ieo.it; 3Division of Epidemiology and Biostatistics, European Institute of Oncology IRCCS, 20141 Milan, Italy; davide.radice@ieo.it; 4Bianalisi Laboratory, 20841 Carate Brianza, Italy; maria.sandri@bianalisi.it; 5Division of Laboratory Medicine, European Institute of Oncology IRCCS, 20141 Milan, Italy; rita.passerini@ieo.it

**Keywords:** CIN2+ recurrence, HPV persistence, HPV genotyping, High-Risk genotypes, multiple HPV infections, HPV 16/18, treatment failure, test-of-cure

## Abstract

**Simple Summary:**

Women diagnosed with cervical intraepithelial neoplasia grade 2 or worse (CIN2+) and treated by excisional procedures remain at high risk for recurrence over time. “Treatment failure” has been reported in up to 23% of women within two years after treatment. The aim of this study was to investigate the impact of HPV same genotype persistence on CIN2+ recurrence. Our findings confirm that HPV same genotype persistence has 30-fold increased odds of developing CIN2+ recurrence (*p* < 0.001), whereas histological grade, glandular crypt involvement, and margin status are not significantly related with treatment failure. Persistence of multiple genotypes and of HPV 16/18 with or without other HR genotypes show a significant impact on relapse free survival. HPV genotyping as “test-of-cure” enables a personalized risk-based management, by identifying women at higher risk of relapse who need intensive follow-up and avoiding risk of over-treatment in women with new HPV genotype infection after surgery.

**Abstract:**

To evaluate the significance of HPV persistence as a predictor for the development of CIN2+ recurrence and the impact of multiple genotypes and of HPV 16/18 on recurrence risk. A prospective cohort observational study was carried out at the European Institute of Oncology, Milan, Italy, from December 2006 to December 2014. A total of 408 women surgically treated by excisional procedure for pre-neoplastic and neoplastic cervical lesions were enrolled. HPV test was performed at baseline and at first follow-up visit planned at 6 ± 3 months after treatment. Two-year cumulative incidences for relapse were estimated and compared by the Gray’s test. Overall, 96 (23.5%) patients were persistent for at least one genotype at three to nine months from baseline and 21 (5.1%) patients relapsed. The two-year cumulative relapse incidence was higher in HPV persistent patients compared to not-persistent (CIF = 27.6%, 95% CI: 16.2–40.2% versus CIF = 1.7%, 95% CI: 0.3–5.8%, *p* < 0.001), in women with persistent multiple infections (CIF = 27.2%, 95% CI: 7.3–52.3%, *p* < 0.001), and with the persistence of at least one genotype between 16 and 18, irrespective of the presence of other HR genotypes (CIF = 32.7%, 95% CI: 17.9–48.3%, *p* < 0.001), but not significantly different from women positive for single infections or any other HR genotype, but not for 16 and 18. The risk of CIN2+ recurrence should not be underestimated when same HPV genotype infection persists after treatment.

## 1. Introduction

Women diagnosed with cervical intraepithelial neoplasia (CIN) or invasive cervical carcinoma (ICC) and conservatively treated by excisional procedures remain at high risk for persistence or recurrence of CIN grade 2 or worse (CIN2+) over time. Persistent or recurrent CIN2+, also known as “treatment failure”, have been reported in up to 23% of women after treatment, and most cases occur within two years after treatment [1,2,3]. Therefore, a careful post-treatment surveillance is mandatory for at least 25 years, to early detect women at high risk of relapse and to avoid anxiety and overtreatment in patients at low risk.

Several risk factors for “treatment failure” have been identified and include patients’ characteristics (age, parity, smoking, and number of sexual partners) and pathological characteristics (lesion size, histological grade, surgical margins, and glandular crypts involvement), but they do not accurately predict CIN2+ recurrence [4,5].

It has been widely proven that persistent Human Papillomavirus (HPV) infection, in particular of High-Risk (HR) genotypes, is the main risk factor for the development of pre-cancerous and cancerous cervical lesions [6,7]. The post-treatment persistence of the same HR HPV genotype responsible for original cervical lesion occur in about one-third of women and may promote disease recurrence during the follow-up period. Indeed, surgical treatment ensures removal of the lesion, but not necessarily viral clearance [8,9].

The most recent guidelines from Italian group for cervical cancer screening (GISCi) and American Society for Colposcopy and Cervical Pathology (ASCCP) have included HR HPV testing at 6 and 12 months, respectively, as the most accurate surveillance strategy to predict “treatment failure” with high sensitivity [10,11].

The aim of this study was to investigate the magnitude and the significance of HPV persistence as a predictor for the development of CIN2+ recurrence. As secondary outcomes, we evaluated the impact of multiple genotypes and of specific HR genotype, especially HPV 16/18, on recurrence risk.

## 2. Materials and Methods

### 2.1. Population

All women affected by precancerous and cancerous cervical lesions and surgically treated by excisional procedures at the Preventive Gynaecology Unit of the European Institute of Oncology, Milan, Italy, from December 2006 to December 2014, were asked to participate in a prospective cohort observational study.

The local Institutional Review Board approved the study protocol and written informed consent for the use of data for scientific purposes was obtained from all subjects prior to treatment.

Patients were included if the following criteria were met: (a) age at diagnosis of 18 years or older; (b) conservative surgical treatment, including loop electrosurgical excision procedure (LEEP) and laser conization, for the removal of persistent CIN1 (not regressing after at least 3 years) or CIN2+; (c) histological confirmation of any grade of CIN, adenocarcinoma in situ (AIS), and ICC, including squamous carcinomas and adenocarcinomas; (d) known HPV status and (e) expressed willingness to undergo follow-up visits at the European Institute of Oncology, Milan. Patients were excluded in case of (a) negative histology; (b) previous history of HPV vaccination; (c) negative HPV test, unknown HPV genotype or only Low Risk (LR) genotypes infection before treatment.

Data regarding principal clinic, laboratory, and pathological characteristics of patients were recorded in a dedicated database.

HPV test was performed at the time of treatment (“baseline”) and as “test-of-cure” at first follow-up visit planned at 6 ± 3 months after treatment. Patients with a first follow-up visit outside the planned 3–9 months range were excluded from the analysis (Figure 1).

HPV same genotype persistence is defined as the post-treatment detection of at least one genotype that was involved in the development of the original cervical lesion and already found at “baseline”.

Recurrence or “treatment failure” is defined as occurrence of high-grade CIN (CIN2+) after treatment that requires re-excisional surgical treatment during follow-up.

All histological diagnoses at baseline and at recurrence, when occurred, were made on surgical tissue, by dedicated gynecological pathologists working at the Pathology Division of our Institute (Division of Pathology, IEO, European Institute of Oncology IRCCS, Milan, Italy). Glandular crypts involvement and status of surgical margins, detailed as ectocervical and endocervical margins, were described.

### 2.2. Human Papillomavirus DNA Detection and Genotyping

A ThinPrep PreservCyt (Hologic, Inc, Bedford, MA, USA) cervical sample was collected in all patients at baseline and at post-treatment visits, to perform Hybrid Capture 2 HR-HPV Test (HC2; Qiagen, Gaithersburg, MD, USA) and Linear Array HPV Genotyping Test (Roche Diagnostics, Pleasanton, CA, USA), in case of positive HC2 test.

The HC2 test is a sandwich capture molecular hybridization assay. It is a signal amplification detection method based on chemiluminescence that detects the following 13 HR-HPV types: HPV 16, 18, 31, 33, 35, 39, 45, 51, 52, 56, 58, 59, and 68. The DNA:RNA hybrids are captured on a microplate, and the emitted light is measured in a luminometer as relative light units (RLUs). Samples are considered as positive if the ratio RLU/cut-off was >1.0 (equivalent to 1.0-pg HPV DNA/mL). All samples with RLU between 1 and 2.5 should be retested as requested in package insert instructions.

The Linear Array test employs biotinylated PGMY09/11 consensus primers to amplify a 450-bp region of the L1 gene, in order to detect the following 37 HPV genotypes: HPV 6, 11, 16, 18, 26, 31, 33, 35, 39, 40, 42, 45, 51, 52, 53, 54, 55, 56, 58, 59, 61, 62, 64, 66, 67, 68, 69, 70, 71, 72, 73 (MM9), 81, 82 (MM4), 83 (MM7), 84 (MM8), and IS39 e CP6108. Thereafter, the denatured polymerase chain reaction products are hybridized to an array strip containing immobilized oligonucleotide probes. The results are visually interpreted by using the provided reference guide according to manufacturer’s protocol by 2 independent operators and the results should be compared to reach the final one.

### 2.3. Statistical Methods

Patients’ characteristics at baseline were cross tabulated with HPV persistence status at 3–9 months and summarized by counts and percentages, age was summarized by median and interquartile range (IQR). Transition probabilities from single and multiple genotypes infection to negative, single, and multiple infection status at 3–9 months from baseline, were estimated as the proportion of patients in each level status with respect to the status at baseline. Categorical variables were compared using the Fisher’s exact test. Between persistence, status age difference was tested using the Wilcoxon two-sample test. HR genotypes distribution at baseline and at the first follow-up visit by single and multiple was summarized by count and percent; transition probabilities were tabulated and plotted as radar-charts. The null-hypothesis that the infection status at baseline do not differ from infection status at first follow-up visit was tested by the Bowker test [12,13]. Relapse free survival was defined as the time from the first follow-up visit (3–9 months from the baseline visit) to the last follow-up visit without signs of relapse. Relapse free patients at the last follow-up visit or at 2 years from the first follow-up, entered the Cumulative Incidence Function (CIF) estimates as censored. Two-year cumulative incidences for relapse were estimated for and plotted against persistence status, infection status, and HPV-16/HPV-18 specific persistence status, respectively. Cumulative incidences were compared by the Gray’s test. All tests were two-tailed and considered significant at the 5% level. All analyses were done using SAS 9.4 (Cary, NC, USA) radar-chart by R Version 3.6.3 (R Core Team (2020)). Exact computation of the Bowker’s test was done using Mathematica 12.1 (Wolfram Research, Inc., Mathematica, Version 12.1, Champaign, IL, USA, (2020)).

## 3. Results

From December 2006 to December 2014, 1530 women affected by precancerous and cancerous cervical lesions were surgically treated by excisional procedures at the Preventive Gynecology Unit of the European Institute of Oncology, inclusion/exclusion criteria were applied and led to a final sample size of 408 (Figure 1).

At three to nine months from baseline, 96 (23.5%) patients were persistent for at least one genotype. Median age at baseline was 40 years (IQR: 35–46). Age, histology, glandular crypts involvement, ectocervical and endocervical margins, as well as overall surgical margins at baseline, were not significantly associated with HPV persistence status after three to nine months. The lowest not-significant variable was histology (*p* = 0.09) (Table 1). The proportion of persistence was significantly higher for patients with multiple infection (32.1%) at baseline compared to patients with single infections (20.3%) (*p* = 0.01). HPV persistence also showed a significantly (*p* = 0.009) monotone inverse relationship with the number of genotypes detected at baseline irrespective of the genotype. Among 96 persistent patients, 60 (20.3%) had only one genotype, 23 (27.7%) had two different genotypes, and 13 (44.8%) had three or more different genotypes, respectively (Table 1).

Among 296 patients with single infection at baseline, at three to nine months, 236 (79.7%) were HPV-free, 55 (18.6%) still had a single infection, and 5 (1.7%) were positive for more than one genotype. Among 112 patients with multiple infection at baseline, 76 (67.9%) were HPV-free after three to nine months, 19 (17.0%) were positive for only one genotype, and 17 (15.2%) for more than one genotype, respectively (*p* < 0.001, Appendix A).

HPV 16 was the most prevalent genotype at baseline in both single (33.5%) and multiple (51.8%) infections, followed by HPV 31 in both single (26.4%) and multiple (29.5%) infections, respectively (Table 2). All other genotypes prevalence ranged from 0 (genotype 66 in single infections at baseline and genotype 68 in single infections both at baseline and at 3–9 months) up to 22.7%, corresponding to five patients with HPV 52 and five patients with HPV 58 in multiple infections, both at follow-up (Table 2).

Prevalence of genotypes 16, 18, 59, 66, and 68 increased at three to nine months from baseline both in single and multiple infections, whereas genotypes 31, 33, 35, and 45 showed a decreasing trend. Other genotypes showed a mixed pattern: proportion decreased in single infections but increased in multiple infections (HPV 51, 52, 58), or increased in single infections but decreased in multiple infections (HPV 39 and 56) from baseline to three to nine months (Table 2 and Figure 2).

Overall, 21 (5.1%) patients relapsed within two years from the first follow-up visit and the overall two-year CIF was 8.6% (95% CI: 5.1–13.2%) (Table 3).

Compared to 2 (0.6%) HPV not-persistent patients, 19 (19.8%) HPV persistent patients relapsed within two years (CIF = 1.7%, 95% CI: 0.3–5.8% versus CIF = 27.6%, 95% CI: 16.2–40.2% respectively, *p* < 0.001) (Figure 3A).

At 2 years cumulative relapse incidences were: 1.7% (95% CI: 0.3–5.8%), 26.2% (95% CI: 14.1–40.2%) and 27.2% (95% CI: 7.3–52.3%) for women that were HPV-negative (not-persistent), with single or multiple infections at the first follow-up visit, respectively (Figure 3B). Gray’s test for cumulative incidence pairwise comparisons was significant for single infection versus HPV negative (*p* < 0.001) and for multiple infection versus HPV negative (*p* < 0.001), but not for single versus multiple infections (*p* = 0.70). The two-year cumulative relapse incidence was highest for women with the persistence of at least one genotype between 16 and 18 at first follow-up visit, irrespective of the presence of other HR genotypes (CIF = 32.7%, 95% CI: 17.9–48.3%), but not significantly different from women positive for any other HR genotype; however, not for 16 and 18 (CIF = 16.1%, 95% CI: 3.8–36.1%, *p* = 0.13). (Table 3 and Figure 3C).

No other variables were significantly associated with the cumulative incidence of relapse (Table 3).

Infection status was not significantly associated with histology (*p* = 0.49 for single versus multiple infections at baseline and *p* = 0.30 for negative versus single versus multiple infections at first follow-up visit, respectively) (Appendix A).

A subgroup analysis on 42 women persistent for HPV 16 only versus 32 persistent patients with only one other HR genotype showed a not significant two-year cumulative relapse incidence difference (CIF = 28.5%, 95% CI: 11.6–48.2% versus CIF = 24.8%, 95% CI: 8.3–45.7%, respectively, *p* = 0.68) (Appendix A).

No multivariable analyses were done due the overall low number (*N* = 21) of events.

## 4. Discussion

CIN2+ treatment showed a favorable long-term outcome with a two-year cumulative incidence of 8.6%. Our findings confirm that persistent HPV infection after treatment is the only significant predictor for the development of CIN2+ recurrence. Whereas histological grade, glandular crypt involvement, and margin status are not significantly related with treatment failure, HPV same genotype persistence has 30-fold increased odds of developing relapse (OR = 29.9, *p* < 0.001).

Moreover, persistence of multiple genotypes and of HPV 16/18 with or without other HR genotypes show a significant impact on relapse free survival. However, there is not a significant increase of cumulative incidence of relapse at two years in persistent HPV 16 women when compared to other persistent HR single genotype patients.

Overall, 5.1% of patients experienced “treatment failure” and this data is consistent with the prevalence of 3.5–12% recently reported by several authors [14,15,16,17]. In our study, we found a persistence rate of HPV infection after treatment of 23.3%, in line with several authors [15,18] and the results of a previous meta-analysis by Rositch et al. [19]. All recurrences except two occur in HPV persistent patients. Previous studies already showed that HPV persistence is a risk factor for “treatment failure” [17,20,21] since viral clearance is significantly associated with efficacy of surgical treatment of CIN2+ [22]. However, the two observed HPV-negative relapses do not imply that HR HPV is not involved in the development of CIN2+ recurrence, since false-negative HPV results may derive from viral integration inside the cell or low levels of HPV DNA viral load. Indeed, integrated viral genomes could not be detected by HPV tests that amplify regions of the L1 gene, such as Linear Array. In addition, low levels of HPV DNA viral load could be due to either sampling errors or the shedding of a few abnormal cells [23].

Despite previous reports [14,15], we did not find a significant difference in age between HPV not-persistent and HPV persistent patients after treatment.

Furthermore, according to our results, histological grade, glandular crypts involvement, and surgical margins are not significantly associated with post-treatment HPV persistence and did not affect relapse free survival, as already sustained by Byun et al. [14]. On the contrary, in a retrospective study, Fernández-Montolí et al. proved that women with involved surgical margins were at higher risk of “treatment failure” (HR = 7.31, *p* = 0.003) than women with clear margins, and the effect was dependent on the type of margin involvement [15]. Indeed, it has been widely established that positive endocervical margins are associated to increased risk of CIN2+ recurrence. Even a meta-analysis by Arbyn et al. [5] confirmed that incomplete excision of CIN leads to five-fold higher risk of “treatment failure”, conversely to women with negative resection margins. However, this meta-analysis highlighted that, despite its significant association with treatment failure, margin status is not an accurate predictor of CIN2+ recurrence, because of heterogeneous sensitivity and low reproducibility of the assessment of resection margins. As a matter of fact, in their systematic review, relapses occurred in women with positive margins in 56% of cases and in women with negative margins in 16% of cases, respectively [5]. Women with clear resection margins could be at risk of relapse because of viral persistence or of multifocal lesions. On the contrary, most women with involved margins would not develop recurrence over time, due to thermal tissue effect on residual cervix and mainly to viral clearance. Therefore, achievement of negative surgical margins should be balanced with potential risks of adverse obstetrical outcomes, especially in young women [24].

Positive HR HPV testing after treatment has been proven to be more sensitive and similarly specific compared with the surgical margins’ involvement. Although recurrences are more frequent in women with both involved margins and positive HR HPV test after treatment, combination of both risk factors does not show a significant improvement of protection against “treatment failure”. Post-treatment HR HPV positivity seems to be the strongest and most accurate predictor of relapse free survival [5,15].

Our results confirmed the clinical impact of HPV genotyping on management and post-treatment surveillance of CIN2+, as previously outlined by a recent systematic review [25]. Detection of persistent HPV infection after treatment should be considered as the main risk factor for the development of recurrent CIN2+.

Persistence of HPV 16 or 18 is associated with a greater risk of “treatment failure”, as widely reported by previous literature [14,16,17,26], probably due to different HPV clearance rate according to genotype after treatment. Indeed, HPV clearance usually occurs within three months of surgery, whereas HPV 16 and 18 are not rapidly cleared [8]. Therefore, women with persistent HPV 16/18 infection could benefit from a more intensive post-surgical follow-up, including immediate colposcopy at three to six months after treatment and closer retesting, in order to earlier identify any relapse. Nevertheless, according to our results, risk of CIN2+ recurrence in case of HPV 16 and/or 18 persistence is not significantly different when compared to other HR genotype persistence. This evidence has an interesting implication, especially when considering that HPV 16 is the most prevalent genotype in both single and multiple infections, followed by HPV 31 and not HPV 18 [27,28].

Furthermore, persistence of multiple infections showed a higher CIN2+ recurrence rate, as already found by Zhang et al. [17]. However, according to our experience, risk of relapse in persistent multiple infections was higher only when compared with HPV negative patients and not with single HPV infected women at follow-up.

Limits of the study include selection bias related to single center analysis and small sample size, with a total 408 patients, of which 96 (23.5%) tested HPV positive after treatment and only 21 (5.1%) experienced CIN2+ recurrence. Moreover, confidence intervals of hazard ratios are wide due to the small number of recurrence events and thus suggesting an uncertain magnitude of the effect.

## 5. Conclusions

The risk of CIN2+ recurrence should not be underestimated when same HPV genotype infection persists after treatment, even in women with negative resection margins. HPV genotyping can be useful to identify women at higher risk of relapse and to avoid risk of over-treatment in women with new HPV genotype infection after surgery. Thus, HPV genotyping as “test-of-cure” enables a personalized risk-based management, by stratifying women who need more appropriate and intensive follow-up strategies and reassuring women cured by surgical treatment.

## Figures and Tables

**Figure 1 cancers-13-03664-f001:**
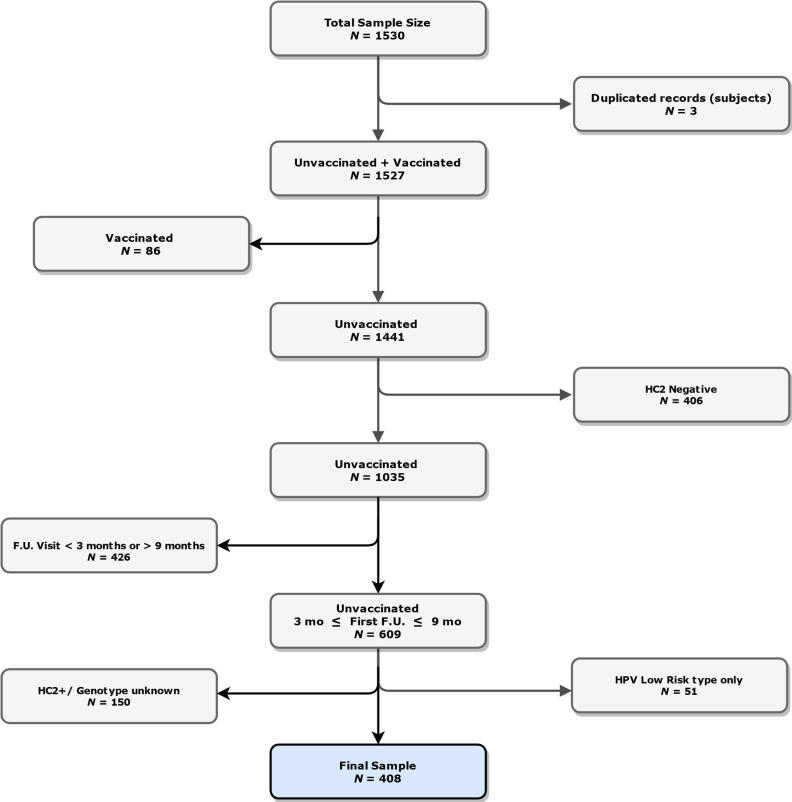
Patients’ inclusion and exclusion criteria flow-chart. HC2 = Hybrid Capture 2, F.U. = follow-up, HPV = Human Papillomavirus, mo = months.

**Figure 2 cancers-13-03664-f002:**
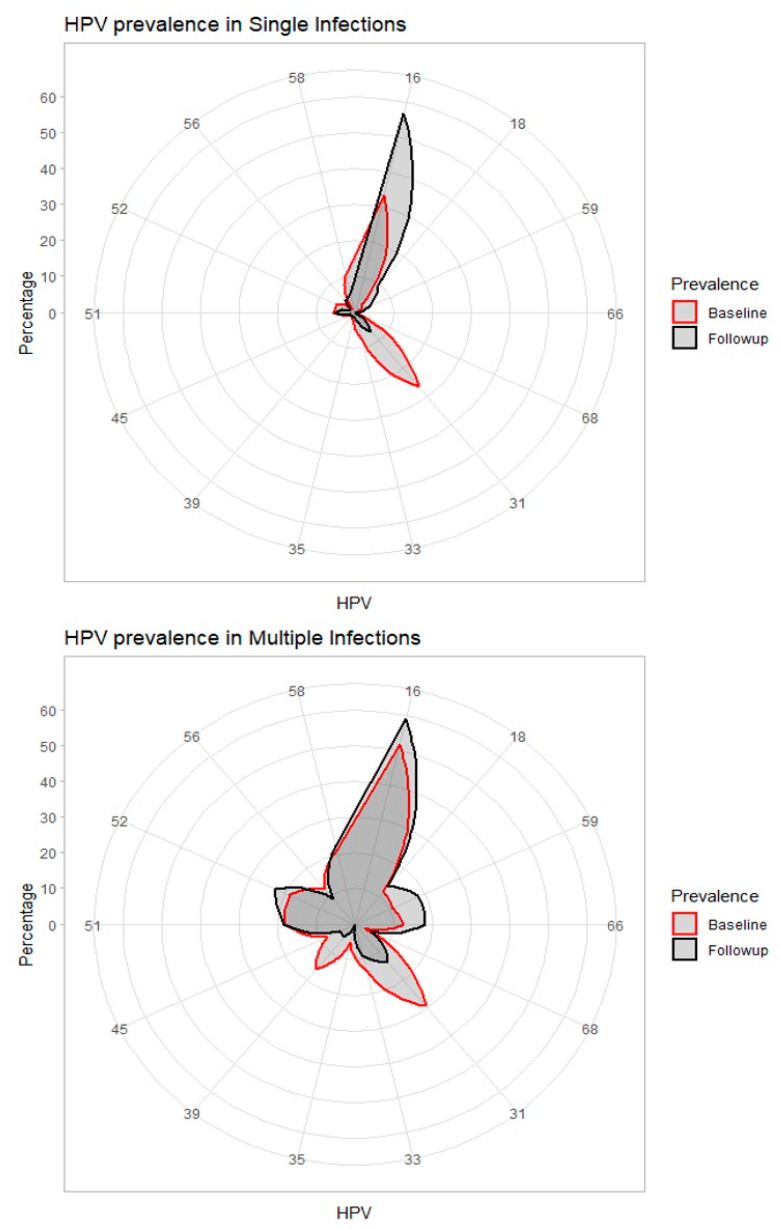
High-Risk Genotypes distribution at baseline and at 3–9 months follow-up visit.

**Figure 3 cancers-13-03664-f003:**
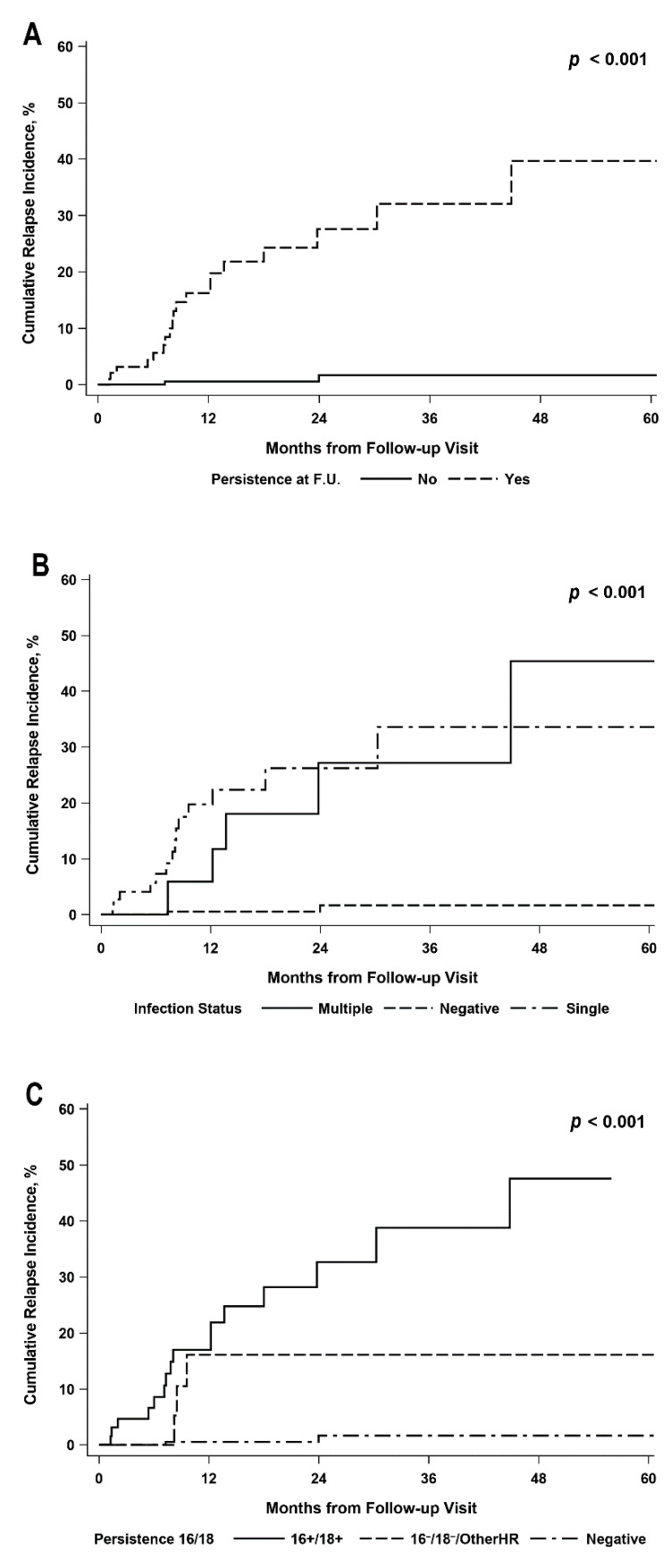
Cumulative relapse incidence by HPV status at first F.U. visit, specifically by HPV infection persistence (**A**), persistence of single or multiple infections (**B**) and persistence of HPV 16/18 or other HR HPV genotypes (**C**). F.U. = follow-up, HR = High-Risk.

**Table 1 cancers-13-03664-t001:** Patients’ characteristics at baseline and HPV persistence status at 3–9 months (first follow-up visit) from baseline.

Characteristic	Level	*N* (Col %)	HPV Persistence, *N* (Row %)		*p*-Value
		All Patients *N* = 408	Not-Persistent *N* = 312	Persistent *N* = 96	
Age, years		40 (35–46) ^a^	39 (35–46) ^a^	40 (35–46) ^a^	0.53
Histology	CIN1	72 (17.7)	59 (81.9)	13 (18.1)	
	CIN2/3/AIS	313 (76.7)	232 (74.1)	81 (25.9)	
	ICC	23 (5.6)	21 (91.3)	2 (8.7)	0.09
Infection status	Single infection	296 (72.6)	236 (79.7)	60 (20.3)	
	Multiple (≥2) infection	112 (27.5)	76 (67.9)	36 (32.1)	0.01
No. of infections ^b^	1	296 (72.6)	236 (79.7)	60 (20.3)	
	2	83 (20.3)	60 (72.3)	23 (27.7)	
	≥3	29 (7.1)	16 (55.2)	13 (44.8)	0.009
Ectocervical margin ^c^	Negative	392 (97.3)	299 (76.3)	93 (23.7)	
	Positive	11 (2.7)	9 (81.8)	2 (18.2)	1.00
Endocervical margin ^c^	Negative	386 (95.8)	298 (77.2)	88 (22.8)	
	Positive	17 (4.2)	10 (58.8)	7 (41.2)	0.14
Surgical margins	Negative	376 (92.2)	290 (77.1)	86 (22.9)	
	Positive	32 (7.8)	22 (68.8)	10 (31.2)	0.28
Glandular crypts involvement	No	192 (47.1)	149 (77.6)	43 (22.4)	
	Yes	216 (52.9)	163 (75.5)	54 (24.5)	0.64

^a^ Median (IQR), ^b^ number of different detected genotypes per patient, ^c^
*N* = 403 (HPV not-persistent *N* = 308, HPV persistent *N* = 95). HPV = Human Papillomavirus, CIN = cervical intraepithelial neoplasia, AIS = adenocarcinoma in situ, ICC = invasive cervical carcinoma.

**Table 2 cancers-13-03664-t002:** High-Risk Genotypes distribution at baseline and at 3–9 months follow-up visit.

Genotype ID	Single *N* (%)		Multiple ^a^ *N* (%)	
	Baseline *N* = 296	Follow-Up ^b^ *N* = 74	Baseline *N* = 112	Follow-Up ^b^ *N* = 22
16	99 (33.5)	42 (56.8)	58 (51.8)	13 (59.1)
18	9 (3.0)	7 (9.5)	13 (11.6)	3 (13.6)
31	78 (26.4)	5 (6.8)	33 (29.5)	3 (13.6)
33	22 (7.4)	2 (2.7)	15 (13.4)	2 (9.1)
35	6 (2.0)	1 (1.4)	6 (5.4)	0
39	3 (1.0)	1 (1.4)	18 (16.1)	1 (4.6)
45	8 (2.7)	1 (1.4)	9 (8.0)	1 (4.6)
51	17 (5.7)	4 (5.4)	20 (17.9)	4 (18.2)
52	15 (5.1)	1 (1.4)	21 (18.8)	5 (22.7)
56	3 (1.0)	3 (4.1)	14 (12.5)	2 (9.1)
58	31 (10.5)	4 (5.4)	24 (21.4)	5 (22.7)
59	5 (1.7)	3 (4.1)	12 (10.7)	4 (18.2)
66	0	1 (1.4)	14 (12.5)	4 (18.2)
68	0	0	3 (2.7)	1 (4.6)

^a^ More than one genotype irrespective of other genotypes, percent do not sum up to 100%; ^b^ 3–9 months from baseline.

**Table 3 cancers-13-03664-t003:** Cumulative Relapse Incidence Functions (CIF) at 2 years by infection status at 3–9 months from baseline and histology.

Characteristic	Level	Events/At Risk	CIF (95% CI)	*p*-Value ^a^
Overall	-	21/408	8.6 (5.2–13.2)	-
Histology	CIN1	1/72	4.2 (0.3–18.0)	
	CIN2/3/AIS	19/313	9.8 (5.7–15.2)	
	ICC	1/23	5.9 (0.3–24.2)	0.21
Persistence	Not-persistent	2/312	1.7 (0.3–5.8)	
	Persistent	19/96	27.6 (16.2–40.2)	<0.001
No. of infections	Negative	2/312	1.7 (0.3–5.8)	
	Single	14/74	26.2 (14.1–40.2)	
	Multiple	5/22	27.2 (7.3–52.3)	<0.001 ^b^
Genotype 16/18 ^c^	Negative	2/312	1.7 (0.3–5.8)	
	16+ or 18+/Other HR	16/64	32.7 (17.9–48.3)	
	16− and 18−/Other HR	3/32	16.1 (3.8–36.2)	<0.001 ^d^
Glandular crypts	No	7/192	6.1 (2.3–12.6)	
	Yes	14/216	11.1 (5.9–18.2)	0.15
Ectocervical margin ^e^	Negative	19/392	8.5 (4.9–13.3)	
	Positive	1/11	11.1 (4.7–40.6)	0.78
Endocervical margin ^e^	Negative	18/386	8.0 (4.6–12.6)	
	Positive	2/17	24.7 (1.5–62.8)	0.20
Surgical margins	Negative	17/376	7.9 (4.4–12.6)	
	Positive	4/32	18.2 (4.6–38.8)	0.09

^a^ Overall Gray’s test, ^b^ Single vs. Multiple, *p* = 0.70; Single vs. Negative, *p* < 0.001; Multiple vs. Negative, *p* < 0.001, ^c^ Single or multiple infections, ^d^ 16+ or 18+/Other HR vs. 16− and 18−/Other HR, *p* = 0.13; 16+ or 18+/Other HR vs. Negative, *p* < 0.001; 16− and 18−/Other HR vs. Negative, *p* < 0.001, ^e^ Based on *N* = 403 valid cases; Median Follow-up: 11.6 months (range: 1.2–78.5). CIN = cervical intraepithelial neoplasia, AIS = adenocarcinoma in situ, ICC = invasive cervical carcinoma, HR = High-Risk.

## Data Availability

The data presented in this study are available on request from the corresponding author. The data are not publicly available due to patients’ privacy restrictions. The data are safely stored in a private database of the European Institute of Oncology, Milan, Italy.

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
