# Peer review of "Human Papillomavirus Same Genotype Persistence and Risk of Cervical Intraepithelial Neoplasia2+ Recurrence"

_cancers, 2021, doi:10.3390/cancers13153664_

Round 1
Reviewer 1 Report
The manuscript entitled 'Human Papillomavirus Same Genotype Persistence and Risk of Cervical Intraepithelial Neoplasia 2+ Recurrence' describes a follow-up study conducted on women recruited and treated for cervical precancer by excisional procedures. This manuscript discusses the risk of developing cervical disease recurrence as 'treatment failure' and evaluates the risk factors involved in the outcome. The follow-up study relies on data collected at one follow-up visit at 3-9 months from treatment ("baseline").
The study was sound and well-conducted with methods properly described and with a solid analysis. Further, the study confirms HPV persistence to be the sole risk factor for cervical disease recurrence post ablation treatment of cervical preneoplastic lesions.
The figures could benefit from additional details in the figure legend.
In the Discussion section, the authors state that the 2 cases of HPV-negative relapse could be related to possible 'false-negative' HPV results that derive from viral integrations into the host cell. However, viral genomes integrated could also be detected by HPV DNA testing. Please elaborate.
Author Response
Thank you for your valuable comments and suggestions. Please find below my point-by-point replies to your concerns.
- The figures could benefit from additional details in the figure legend.I thank you for the suggestion and I added more details in figure and table legends.
- In the Discussion section, the authors state that the 2 cases of HPV-negative relapse could be related to possible 'false-negative' HPV results that derive from viral integrations into the host cell. However, viral genomes integrated could also be detected by HPV DNA testing. Please elaborate.I agree with your comment and I edited the paragraph at lines 246-252 in the Discussion section, in order to better explain this concept as suggested.
Reviewer 2 Report
The paper proposed by A D Iacobone et al. and entitled “Human Papillomavirus Same Genotype Persistence and Risk of Cervical Intraepithelial Neoplasia 2+ Recurrence” reports the result of a prospective study designed to assess the relative risk of relapse of high grade CIN according to persistence of viral infection after removal of the lesion. The study is based on the analysis of a case panel of four hundred eight women surgically treated by excisional procedure for pre-neoplastic cervical lesions. HPV test was performed at the time of treatment and at the first follow-up visit planned at 6 ± 3 months after treatment. Recurrence was defined as occurrence of high-grade CIN (CIN2+) after treatment. Viral analysis was performed using Hybrid Capture 2 (HC2) test and Linear Array HPV Genotyping test in case of positive HC2 test. At 3-9 months from baseline, 96 (23.5%) patients were persistent for at least one genotype. Of these, 19 (19.8%) relapsed within 2 years compared to 2 (0.6%) HPV not-persistent patients, ( p < 0.001). Further analysis showed that the 2-years cumulative relapse incidence was highest for women with the persistence of HPV16 and/or HPV18 genotypes at first follow-up visit, irrespective of the presence of other HR genotypes, but not significantly different from 207 women positive for any other HR genotype but not for 16 and 18.
On the whole, this study shows that the persistence of HPV16/18 genotype is a risk factor for relapse of high grade CIN (16/64), but that the relapse incidence is low in patients positive for any other genotype (3/32) or that remain HPV negative (2/312).
The study is well performed, the methodology sound and the result significant. The main limitation is that, in clinical practice, although the author state that “women with persistent HPV 16/18 infection could benefit from a more intensive post-surgical follow-up”, they do not detail what improvement in follow-up could be proposed and what could be the benefit for patients as compared with the “standard follow-up”. This point could be briefly developed. Nevertheless, “viral test of cure” that allows reassuring women after surgery can be useful.
Author Response
Thank you for your pleasing comments.
As suggested, at lines 290-291 I briefly explained what improvement in follow-up could be proposed and what could be the benefit for women with persistent HPV 16/18 infection in clinical practice.